# CIS²: A Simplified Commonsense Inference Evaluation for Story Prose

**Bryan Li, Lara J. Martin,** and **Chris Callison-Burch**
University of Pennsylvania
Philadelphia, PA, USA
`{bryanli, laramar, ccb}@seas.upenn.edu`

## Abstract

*Contextual Commonsense Inference (CCI)* is the problem of inferring causal relations between the events of a text, such as a story. Like other commonsense reasoning tasks, CCI is a problem of language understanding, rather than language generation. We show that prior work, in using language generation to perform CCI, trains models that struggle on the CCI task in isolation. This *conflation* of tasks is further exacerbated by evaluating with word-matching based metrics such as BLEU. In order to isolate CCI from language generation, we reframe CCI as a classification problem. Our system, which we call CIS², forces the model to focus on CCI directly by providing it the original text of the story to use for understanding while having it generate only the bare minimum: indices to sentences. We look at the GLUCOSE (Mostafazadeh et al., 2020) dataset and compare against their task for predicting CCI between story sentences. We find that models trained on CIS² index labels achieve a 4.3% higher CCI accuracy than those trained for generating full phrases, such as in the original GLUCOSE task.

## 1 Introduction

Transformer-based language models (Vaswani et al., 2017)—particularly off-the-shelf models—have shown mixed success with story generation (See et al., 2019; Wang and Wan, 2019; Ippolito et al., 2020). Language models (LMs) lose coherence as their output length increases, and are prone to meandering, losing the plot of a story over time. This can be largely attributed to the LM generating each token by sampling from a probability distribution, failing to distinguish between statistical correlation (how frequently event A and event B are seen together) and causal reasoning (event A causes event B to occur).

Since causal events across sentences in stories help people understand and retain story information

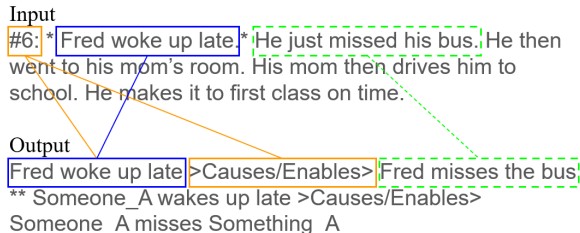

Figure 1: Motivation for CIS², illustrating how the original GLUCOSE task conflates commonsense inference and text generation. Input and output are exactly as seen by finetuned T5. Blue: selected sentence *X* is always paraphrased. Orange: dimension specifies the position of *X*, and the relation. Green: commonsense inference is needed here to select the other sentence *Y*.

(Trabasso et al., 1984), we posit that the inability of language models to perform commonsense inference leads them to output less coherent long-form text. Commonsense inference is still an open problem in NLP, especially when the commonsense information is unstructured and provided in the form of natural language. We refer to this task of grounding commonsense inference relations within prose as *contextual commonsense inference (CCI)*, a sub-task within commonsense reasoning. Due to storytelling being deeply intertwined with causal understanding, improving CCI will yield both more accurate story generation evaluation metrics and better story generation.

Current methods in CCI for story understanding often include the use of generative LMs. While LMs might be helpful for encoding the textual information, they are less suited to operating on and making decisions based on this information due to their probabilistic way of generating text. This leads to a tendency to focus on grammar rather than meaning (Martin et al., 2018). Furthermore, commonly-used language generation evaluation metrics like BLEU put emphasis on exact word usage and grammar. In this paper, we look at what it would mean to de-emphasize generation and para-

phrasing for understanding tasks like CCI.

Our contributions in this paper are twofold. First, we critique an existing method addressing the *contextual commonsense inference* (CCI) task by using the GLUCOSE (Mostafazadeh et al., 2020) dataset and teasing apart their associated CCI task formulation. We designed several diagnostic tasks which selectively omit sentences of the input and investigate which sentences contribute the most to paraphrasing/generation. We replicate their results, then finetune T5 models (Raffel et al., 2020) on each of our diagnostic tasks, to show the significant conflation of language understanding and generation in the original GLUCOSE T5 model.

Second, we propose $\text{CIS}^2$ (Contextual Commonsense Inference in Sentence Selection), a simplified task for more fairly evaluating commonsense inference in storytelling, which abstracts away the natural language generation component almost entirely. We develop a heuristic to convert story sentences into $\text{CIS}^2$ tags and show that a language model, when trained on this data, outperforms the original GLUCOSE task formulation on forming the correct causal relations between sentences in stories. Our findings reinforce that while the GLUCOSE dataset encodes useful commonsense information, we urge that future work should carefully disentangle language generation when performing language understanding tasks.

## 2 Related Work

Commonsense inference is the ability to use prior knowledge based on real world experiences to infer what has happened or will happen. While lived experiences vary from person to person, there are still significant commonalities as we live and interact within the same physically- and temporally-constrained world.

### 2.1 Commonsense Knowledge Graphs

Hwang et al. (2021) formalized the *commonsense inference task* (CI) for AI systems as a knowledge three-tuple, to predict the *object* of a relation given the *subject* and *relation*. This formulation of commonsense inference can be structured as a graph, where the subjects and objects are nodes and the relations are the edges connecting the entities. These commonsense knowledge graphs (CKGs) explicitly encode the structure of inference relationships between entities. ATOMIC (Sap et al., 2019) is one such CKG dataset that organizes everyday events

into if-then relationships. COMET (Bosselut et al., 2019) is a transformer language model designed on top of ATOMIC relations, showing language models can encode and generalize commonsense information.

However, Wang et al. (2021) show that language models struggle to perform generalizable commonsense inference across three popular CKG datasets: ConceptNet (Speer et al., 2017), TupleKB (Dalvi Mishra et al., 2017), and ATOMIC (Sap et al., 2019). They found that LMs trained on several CKGs have limited ability to transfer knowledge to unseen CKGs, and that adaptation generalizes well to unseen subjects, but less so on unseen objects.

Although these graphs do well at representing facts and their relations, their statements lack context and would need to be adapted to a textual domain, such as story prose. Using them to generate a story as-is would fail to engage readers since the "story" would simply be a series of facts. Our work goes beyond the explicit structure of CKGs, focusing on finding and leveraging commonsense relations in natural language short stories.

### 2.2 Commonsense Inference for Storytelling

Early research on automated story generation research focused on designing systems that create *coherent* stories (Lebowitz, 1985; Turner and Dyer, 1985; Liu and Singh, 2002; Young et al., 2013). Despite the success of neural networks for AI tasks, commonsense and coherence remain big issues for story generation systems.

Applying commonsense reasoning to the events of a story has been proposed as one way to tackle the difficult problem of assessing the quality of machine-generated stories. The Story Cloze Test (Mostafazadeh et al., 2016) formulates story ending generation as a multiple-choice task, having systems look at several possible endings and predict the one that is most reasonable. Guan et al. (2019) integrated commonsense reasoning directly into their Story Cloze model by building context clues and using implicit knowledge.

Commonsense reasoning can also help story generation with issues in plot coherence. Martin (2021) created a neurosymbolic system that leveraged VerbNet (Brown et al., 2019) facts to ground neural story generation in commonsense reasoning. They did this by tracking the story state and pruning out impossible options that a neural network

| # | Description | Relation Text |
|---|---|---|
| 1 | Event that causes or enables X | >Causes/Enables> |
| 2 | Emotion/basic human drive that motivates X | >Motivates> |
| 3 | Location state that enables X | >Enables> |
| 4 | Possession state that enables X | >Enables> |
| 5 | Other attributes enabling X | >Enables> |
| 6 | Event that X causes or enables | >Causes/Enables> |
| 7 | An emotion that is caused by X | >Causes> |
| 8 | A change in location that X results in | >Results in> |
| 9 | A change of possession that X results in | >Results in> |
| 10 | Other changes in property that X results in | >Results in> |

Table 1: The ten GLUCOSE dimensions and the corresponding relation text connecting statements (Mostafazadeh et al., 2020).

| Parameter | Text |
|---|---|
| Story | Fred woke up late. He just missed his bus. He then went to his mom's room. His mom then drives him to school. He makes it to first class on time. |
| Selected Sentence ($X$) | Fred woke up late. |
| Dimension | 6 |
| Specific Rule | Fred wakes up late >Causes/Enables> Fred misses his bus |
| General Rule | $Someone_A$ wakes up late >Causes/Enables> $Someone_A$ misses $Something_A$ |

Table 2: Example GLUCOSE entry (Mostafazadeh et al., 2020). The top three rows (story, $X$, dimension) are input, and the bottom two rows (specific rule, general rule) are output.

provided as candidate next sentences for the story. Similarly, the Commonsense inference Augmented neural StoryTelling (CAST) framework (Peng et al., 2021) modeled interactions between multiple characters using ATOMIC. The stricter, more explicit generation constraints of CAST produced more coherent and on-topic two-character stories than generating via sampling from a distribution alone.

TellMeWhy (Lal et al., 2021) is a dataset built on top of ROCStories (Mostafazadeh et al., 2016), consisting of 30k questions on why characters perform their actions and the corresponding answers. They found that current state-of-the-art models performed far worse than humans, especially on questions whose answers are external to the narratives. This contrasts with the findings discussed in Mostafazadeh et al. (2020) that language models can approach human performance.

## 3 The GLUCOSE Dataset and Task

Our work follows from GLUCOSE (GeneraLized and COntextualized Story Explanations) (Mostafazadeh et al., 2020). In this section we briefly describe their dataset and experiments; for more details, refer to the original paper. The GLUCOSE dataset contains 670K crowdsourced annotations identifying causal reasoning relations between the sentences within stories from ROCStories (Mostafazadeh et al., 2016)—a collection of crowdsourced five-sentence everyday stories in English. The authors structured the collected data around ten different dimensions, shown in Table 1, of causal relations between a pre-selected sentence $X$ from the story and another statement $Y$, which can either be another story sentence or some ex-

ternal commonsense knowledge. The relationship between these statements can be formalized as:

$$statement_1 \; REL \; statement_2 \qquad (1)$$

$X$ can be in either *statement* position, depending on the particular dimension chosen: Dimensions 1-5, specify events that *caused* $X$ (i.e., $X$ is *statement$_2$*), and dimensions 6-10 specify events *caused by* $X$ (i.e., $X$ is *statement$_1$*).

### 3.1 Contextual Commonsense Inference Task

GLUCOSE addresses the task of predicting relationships between statements explicitly or implicitly expressed within a text, a task we term *contextual commonsense inference* (CCI). An example GLUCOSE entry can be found in Table 2. The entries are organized to reflect the CCI task and are formalized as input-output tuple pairs, with input tuple

$$\langle \text{story } S, \text{ selected sentence } X, \text{ dimension } D \rangle, \quad (2)$$

where a story $S$ consists of five sentences [$s_0$, $s_1$, $s_2$, $s_3$, $s_4$], the selected sentence $X$ is the sentence on which the rule is centered, and the number dimension $D$ is one of the ten dimensions from Table 1—and output tuple

$$\langle \text{specific rule } R_S, \text{ general rule } R_G \rangle, \quad (3)$$

where the specific rule $R_S$ is the relation between $X$ and $Y$. $Y$ can be either (1) another sentence in the story or (2) an implicit statement from outside the text. The general rule $R_G$ is the same rule as $R_S$ but using generalized tags for named entities

| Task | Input | Output |
|------|-------|--------|
| ORIGINAL | 1: My mother told me to fix the car. I was unable to do this right away. **\* I could not find my tools. \*** I looked everywhere for them. It turns out they were stolen the night before. | They were stolen the night before >Causes/Enables> **I could not find my tools** \*\* Something$_A$ is stolen >Causes/Enables> Someone$_A$ cannot find Something$_A$ |
| HISTORY | 1: My mother told me to fix the car. I was unable to do this right away. | They were stolen the night before >Causes/Enables> **I could not find my tools** \*\* Something$_A$ is stolen >Causes/Enables> Someone$_A$ cannot find Something$_A$ |
| MASK X | My mother told me to fix the car. I was unable to do this right away. `<masked>` I looked everywhere for them. It turns out they were stolen the night before. | They were stolen the night before >Causes/Enables> **I could not find my tools** \*\* Something$_A$ is stolen >Causes/Enables> Someone$_A$ cannot find Something$_A$ |
| HISTORY+X | 1: My mother told me to fix the car. I was unable to do this right away. **\* I could not find my tools. \*** | They were stolen the night before >Causes/Enables> **I could not find my tools** \*\* Something$_A$ is stolen >Causes/Enables> Someone$_A$ cannot find Something$_A$ |
| CIS[2] | 1: My mother told me to fix the car. I was unable to do this right away. **\* I could not find my tools. \*** I looked everywhere for them. It turns out they were stolen the night before. | `<s_4>` `>Causes/Enables>` `<s_2>` |

Table 3: Task formulations of the same GLUCOSE entry. The output is split into a specific rule and a general rule by "\*\*", and the selected sentence $X$ ("I could not find my tools") is surrounded by single asterisks. In this table, we also **bolded** the selected sentence, and special tokens are `monospace`. The "1:" at the beginning of the input specifies the GLUCOSE dimension; "1" corresponds to the Causes/Enables relation. The diagnostic tasks HISTORY, MASK X, and HISTORY+X are variations on the original task, ORIGINAL. CIS[2] is our proposed task.

(e.g., Someone$_A$ instead of Fred). To summarize, the GLUCOSE task is: given *S, X, and D*, predict/generate $R_S$ and $R_G$.

In this paper, we compare to their best model, a finetuned T5 model (Raffel et al., 2020), which achieved a 71.26 average SacreBLEU (Post, 2018) across the 10 dimensions on predicting general rules and a 75.65 average for the specific rules.[1] The models were also rated for "correctness" using crowdsourcing, where their T5 model scored 2.5/3 averaged across all 10 dimensions on a 4-point Likert scale mapped to a numerical scale of 0-3. For context, their closest baseline got a 2.21/3 average and the gold standard was 2.8/3.

### 3.2 Issues with the GLUCOSE Task for CCI

We find that the GLUCOSE dataset is well-designed and of good annotation quality. However, we take issue with the GLUCOSE task, which asks a model to perform two tasks simultaneously: commonsense inference and language generation. Due to this *conflation* of tasks, the model, in generating its output, would rely heavily on the already-good language generation ability of T5 and neglect learning enough CCI. T5 (Raffel et al., 2020) and other

---

[1]Our best-effort replication of their experiments achieves slightly lower BLEU scores (66.2 & 70.7, respectively) due to resource limitations (detailed in Appendix A.4).

transformer LMs were designed to perform language *generation* tasks. Therefore, by including text generation as part of CCI, T5 will focus on paraphrasing or even copying story sentences.

There are several one-to-one correspondences between parts of the input and output in the original GLUCOSE task (illustrated in Figure 1). For example, for all GLUCOSE entries, the output contains at least one paraphrased sentence from the input. Conflation with paraphrasing worsens with BLEU as the evaluation metric, where incorrect commonsense inferences can score partial credit if they have words in common.

## 4 Diagnostic Tests

In this section, we describe our three diagnostic tests—variations on the original GLUCOSE task with altered input—to isolate different factors that influence T5's generation. Through these tests, we investigate the extent to which language models rely on paraphrasing to generate the commonsense rule output for GLUCOSE.

For each of the following diagnostic tests, we finetune the same T5 (Raffel et al., 2020) model, a pretrained model using the same hyperparameters as in the GLUCOSE paper, to generate the same output as in Equation 3. The diagnostic tests differ only in the format of the input. The purpose of

these tests was to assess how reliant the model is on language generation when performing CCI. More detailed training setup and hyperparameters for these models can be found in Appendix A.5.

Because these tasks are measured with BLEU, conflation between CCI and language generation will always occur. But by deleting different parts of the input, these diagnostic tasks analyze which sentences contribute the most to performance, thus resulting in more conflation.

An overview of the tests' different data formats can be found in rows 2, 3, and 4 of Table 3. We describe them in this section using the following terminology for brevity:
*Dimension (dim)*: the causal dimension
*Pre-context*: sentences before selected sentence X
*Selected sentence (X)*: the story sentence of interest
*Post-context*: sentences after selected sentence X

**ORIGINAL.** This experiment is the same as in (Mostafazadeh et al., 2020), which we described in Section 3.1. We report results on our own replication of the finetuned T5 model, implemented with the `transformers` package (Wolf et al., 2019).

**HISTORY.** This experiment gives as input only the pre-context (the sentences before sentence $X$) and the dimension. This model must generate the output without knowing the target sentence $X$, nor the events happening afterwards. Here, we test the model's ability to generate two (specific) statements given only what happened before. This difficult task serves as a lower bound to contextual commonsense inference performance. Conflation with language generation is absent.

For all dimensions, the model must first speculate what $X$ might be given the pre-context. Based on this predicted X, it generates a statement $Y$ that follows from the causal relationship: either a paraphrase from the input or an implied statement.

**Masked Selected Sentence (MASK X).** This experiment gives as input the pre-context, post-context, and the dimension. The selected sentence is replaced with a token `<masked>`. Here, we test the commonsense ability to generate two (specific) statements given most of the story—4 out of 5 sentences—but not the selected sentence $X$. This will let us see how much of a performance boost the model is given by copying $X$ from the input.

As with HISTORY, for all dimensions, the model must first predict $X$, then generate a paraphrased or implied statement $Y$ that is causally consistent.

| model | spec | spec1-5 | spec6-10 | gen | gen1-5 | gen6-10 |
|---|---|---|---|---|---|---|
| ORIGINAL | 70.7 | 67.1 | 74.4 | 66.2 | 62.3 | 70.0 |
| HISTORY | 35.9 | 36.9 | 34.9 | 50.4 | 50.1 | 50.7 |
| MASK X | 41.6 | 38.8 | 44.4 | 49.6 | 50.4 | 48.8 |
| HISTORY+X | 68.3 | 66.2 | 70.4 | 65.5 | 61.8 | 69.3 |

Table 4: Test SacreBLEU scores for the diagnostic tasks. ORIGINAL performs the best since it can access the entire input. As we keep the output and underlying T5 LM consistent but vary the input, the results' trends demonstrate how omitting different parts of the input affect BLEU scores.

**History and Selected Sentence (HISTORY+X).** This experiment gives as input the pre-context, selected sentence, and dimension. This is used as a direct comparison to HISTORY except with selected sentence $X$ given as part of the input. Statement $Y$ is generated as it is in HISTORY.

For this diagnostic test, we drop entries in which the modifications result in input identical to the original task. For example, for HISTORY+X, we omit those entries where $X$ is the last sentence.

## 4.1 Diagnostic Task Results

Table 4 compares the results of T5 models trained on the diagnostic tasks. We report test set results on the averaged dimensions 1-10, as well as averaged dimensions 1-5 ($X$ is the second statement), and 6-10 ($X$ is the first). Following Mostafazadeh et al. (2020), we use SacreBLEU (Post, 2018) with equal weights up to 4-grams. We report results for both specific and general rules, but focus on specific.

ORIGINAL, of course, performs the best as its input has the most available information. HISTORY and MASK X perform similarly to each other and far worse than the other diagnostic tasks. HISTORY, with only the pre-context, has a a 35-point BLEU gap for specific rules (16 for general) compared to ORIGINAL averaged across all dimensions.

Adding to HISTORY multiple sentences of the post-context gives MASK X, and modest score gains (35.9 vs 41.6 specific). However, adding to HISTORY just the one selected sentence $X$ gives HISTORY+X, which performs very closely to ORIGINAL for both specific and general rules (70.7 vs 68.3 specific). Furthermore, comparing trends between dimensions 1-5 and 6-10, we find that 6-10 scores are mostly higher, for both general and specific, than 1-5.

These results and their trends show that BLEU scores are highly contingent on having $X$ as input over all other sentences. Conflation always occurs

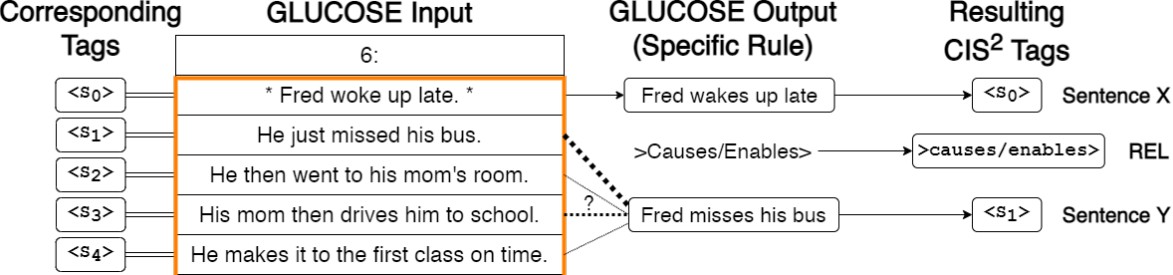

Figure 2: Generation of CIS² labels from a GLUCOSE entry. The input story is highlighted in orange. Each story sentence is indexed by its position in the story. For example, the selected sentence *X* (*Fred woke up late.*), surrounded with asterisks, is assigned the tag `<s_0>`. The relation `>Causes/Enables>` is given automatically from the dimension. The "other" sentence *Y* is compared to each story sentence; the dashed lines represent sentence similarity scores, with the darkest line being the highest similarity. `<s_1>` is selected as the Sentence *Y* tag.

for *X*, since this is copied from the input, and conflation is also worse in cases where an incorrect statement *Y* was generated but contains tokens that match the correct statement. We believe it is unlikely that achieving ~35.9 BLEU on specific rules for HISTORY would mean that it is half as good at CCI than ORIGINAL, with 70.7 BLEU specific. We found that the fine-tuned T5 models perform some CCI, but BLEU scores are hard to interpret and can be unreliable.

**Specific vs. General Rule Performance** Table 4 shows that both ORIGINAL and HISTORY+X perform better for specific rules than general. This matches the results seen in (Mostafazadeh et al., 2020). However, for HISTORY and MASK X, which both omit *X*, the opposite trend occurs; general is higher than specific. This shows that copying and paraphrasing from the original text is in fact a conflating factor in the LM's BLEU performance.

## 5 Contextual Commonsense Inference in Sentence Selection (CIS²)

Given the extensive paraphrasing present in both the GLUCOSE task and the evaluation method, we design the Contextual Commonsense Inference in Sentence Selection (CIS²) task to abstract away language generation. We recast the task as a classification problem, with the same 3 inputs as in ORIGINAL (Equation 2), while the output becomes

$$\langle \texttt{<s}_\texttt{a}\texttt{>} \ \text{REL} \ \texttt{<s}_\texttt{b}\texttt{>}\rangle \tag{4}$$

where `<s_a>` and `<s_b>` are tags corresponding to sentences from the original story, $a$ and $b$ are indices from $[0, 4]$ and $a \neq b$. The output sequence comes from a limited vocabulary of 5 sentence in-

dex tokens, 5 causal dimension tokens,[2] and the sentence index token corresponding to the selected sentence *X* can be before or after the REL token, depending on what causal dimension is being used. The classification task is to choose the correct sequence of 100 possible output sequences.[3]

The abstracted output avoids the prior conflation issue since there are no partial matches within tokens of statements. Furthermore, there is no explicit correspondence between input and output. Note that CIS² does not distinguish between specific and general rules.

Finetuned CIS² models are forced to only learn the commonsense inference task. The input is kept the same, so the models see the same information as with the original task formulation. Therefore, we argue that CIS² is a simpler and fairer measurement of commonsense inference performance.

### 5.1 GLUCOSE Entries to CIS² Tag Heuristic Conversion

To evaluate the CIS² formulation, we need to convert story sentences into CIS² output labels, as in Equation 4. See Figure 2 for the conversion process. Each sentence of an input story corresponds to a tag `<s_0>` to `<s_4>` with indexes corresponding its position in the story. To get the three CIS² output labels, we do the following: (1) Identify selected sentence *X* from the input since it always be denoted as the sentence with the asterisks surrounding it. The input dimension informs the position of sentence *X* in the output—whether is `<s_a>` or `<s_b>`; (2) Get the relation REL from the output directly; and (3) Calculate the similarity of "other"

---

[2] `>Causes/Enables>`, `>Causes>`, `>Enables>`, `>Results in>`,`>Motivates>`

[3] 20 (5P2) sentence tag combinations * 5 relations = 100

sentence $Y$ from the output to every other sentence in the input story and select the closest match.

To find the remaining token, we look at the specific rule from the original GLUCOSE task output, which consists of two statements separated by relation `REL`. We will call them $P_0$ and $P_1$. Suppose $X$ corresponds to $P_0$, and we need to find which sentence $Y$ corresponds to $P_1$. We do this by iterating over the sentences (excluding X), for each calculating its similarity with $P_1$. We take the index of the sentence with the highest similarity to $P_1$ as `<s_b>`. We describe our experiments with several sentence similarity metrics in Section 5.2.

Being a heuristic approach, generated $CIS^2$ labels are not perfect. However, our manual inspection finds most labels are reasonable for GLUCOSE entries that have an explicit $Y$ (from the story). $CIS^2$ labels do not exist for those GLUCOSE entries with implicit relationships[4], i.e. $Y$ is not in the original story. We attempted to filter these out by removing any training examples that did not pass a threshold[5] of SBERT $\leq 0.16$ for any sentence in the story. However, this resulted in a slight drop in the final evaluation, so these examples were kept.

We run the conversion method on the GLUCOSE train set and train a T5 model using the same hyperparameters used for our other models with the task of generating the three-token $CIS^2$ label, given the GLUCOSE input. We refer to this model as $CIS^2$-T5. Note that although using $CIS^2$ tags turns this into a classification problem, the model is still doing generation to predict the output.

## 5.2   $CIS^2$ Classification Task & Results

In Section 4, we showed that BLEU is not an appropriate metric for the CCI task, given the GLUCOSE models' extensive copying and paraphrasing. Furthermore, $CIS^2$-T5 generates $CIS^2$ tags instead of full sentences, making it non-trivial to compare to the ORIGINAL GLUCOSE T5 model.

We run the conversion method from Section 5.1 on each model's specific rule output to obtain its predicted $CIS^2$ labels, and on the GLUCOSE test set to obtain the $CIS^2$ test set.[6] Both are now formatted as in Equation 4. This enables us to do an exact-match comparison between the model labels and the test set labels, and removes the associated issues with evaluating generated text. In effect, the

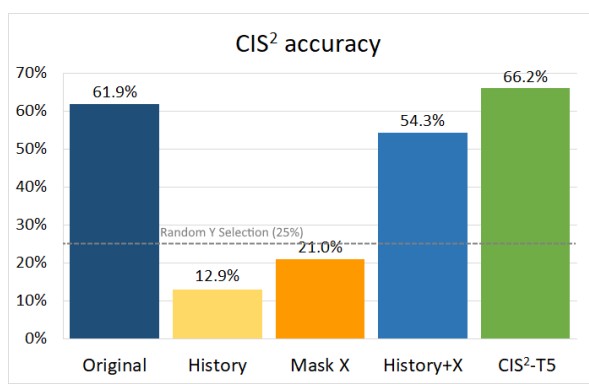

Figure 3: $CIS^2$ accuracy results for Original and diagnostic GLUCOSE task models, and $CIS^2$-T5. The dashed line shows Random Y Selection, a baseline that derives $X$ and the relation text from the input, and randomly selects $Y$.

$CIS^2$ evaluation considers requires *the correct sentence Y to be chosen*; there is no partial credit for those outputs that can easily be inferred from input: the selected sentence $X$, and `REL`.

The sentence similarity metric used is crucial in the process of heuristically generating $CIS^2$ labels. We experimented with both BLEU scores of lemmatized tokens, as well as Sentence-BERT (SBERT) (Reimers and Gurevych, 2019). By using BLEU for sentence similarity, GLUCOSE ORIGINAL achieves 66.0%, whereas $CIS^2$-T5—despite being trained on these $CIS^2$ labels converted with BLEU—only achieves 57.2% accuracy. This stems from same issues of BLEU measuring language generation, rather than CCI, as discussed in Section 4. Also, this shows that the $CIS^2$ classification task does not favor our $CIS^2$ system by default.

Therefore, for the final evaluation we opt for SBERT, a more context-dependent similarity metric. Results for this evaluation are shown in Figure 3. We compare all of our results to a random baseline which is the probability one of the 4 other story sentences is randomly selected for the index of $Y$; this would have an accuracy of 25% (the dashed horizontal line in Figure 3). Out of all the models, $CIS^2$-T5 achieves the highest score at 66.2%, while ORIGINAL is not far behind at 61.9%. As for the diagnostic tasks, we see the same score ordering of models with BLEU evaluation. HISTORY+X scores 8% lower than ORIGINAL. HISTORY and MASK X perform even worse than random, indicating that their BLEU performance was

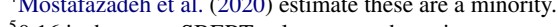

[4] Mostafazadeh et al. (2020) estimate these are a minority.

[5] 0.16 is the mean SBERT value across the train set.

[6] For future work we plan to obtain ground-truth test labels via crowdsourcing.

largely due to partial token matches.[7]

The best GLUCOSE model ORIGINAL achieves 70.7 specific BLEU, but only 61.9% CIS$^2$ accuracy. Although we cannot directly compare BLEU of generated output, and CIS$^2$ exact match accuracy, we have shown that CIS$^2$ provides a fairer estimate of CCI performance of these fine-tuned T5 models by removing language generation from evaluation. These CIS$^2$ results are promising, but there is still much room for improvement.

## 6 Discussion

The diagnostic tasks we discussed in the paper investigated the extent to which the original GLUCOSE task conflates language generation and contextual commonsense inference (CCI). We found that the most significant sentence of the input is the selected sentence *X*, and if omitted, BLEU scores drop significantly compared to omitting other story sentences. This shows that the language model is relying on *X* for CCI, as it should. It is worth discussing how "fair" it is to remove *X*—after all, without *X*, the LMs have little to condition their predictions on. While this is true, we emphasize that our diagnostic tasks are intended to be taken together to analyze the extent of conflation. The main takeaway is that by including *X*, trained models will rely on copying instead of good commonsense inference.

We have also shown evidence for extensive copying and paraphrasing as seen from the higher performance on specific rules relative to general rules for ORIGINAL and HISTORY+X. These trends hold for CIS$^2$ evaluation as well, but are even more marked since there is no inflation from matching tokens.

Lastly, we have shown that the T5 model trained on the GLUCOSE task (to maximize BLEU on the specific and general rules) performs only 4.3% worse on the CIS$^2$ than one trained directly on CIS$^2$ labels. This shows that T5 can still learn significant CCI from the GLUCOSE data, and can further improve performance with CIS$^2$ converted labels, abstracting away with language generation.

### 6.1 Future Work

We plan to collect ground-truth CIS$^2$ labels via crowdsourcing for the entire test set, and for some training examples. To simplify the task, we will have workers verify, and correct if necessary, the heuristic CIS$^2$ labels.

Future work can further explore utilizing GLUCOSE and related datasets for story generation tasks. One promising avenue to extending our CCI evaluation to story generation settings is incorporating our approach with the COINS framework (Paul and Frank, 2021), which generates contextualized inference rules to guide future output sentences. Abstracting these inference rules through CIS$^2$ would likely allow the language model to better capture and learn CCI.

We also resonate with question-answering based approaches to commonsense inference for stories (Lal et al., 2021; Castricato et al., 2022). Lal et al. (2021) trained large language models on their dataset, finding that they only perform well when the answers are present in the narrative. This finding goes hand in hand with our finding that the original GLUCOSE task formulation allows for easy paraphrasing and thus inflated performance.

## 7 Conclusion

This work investigated the extent to which language models learn contextual commonsense inference (CCI), utilizing the GLUCOSE (Mostafazadeh et al., 2020) dataset and the T5 (Raffel et al., 2020) language model as case studies. We showed how the original GLUCOSE task conflates language generation and CCI tasks, causing over-estimation of true CCI performance. We then formulated diagnostic tasks by permuting the original task and found that LMs rely on paraphrasing the selected sentence and context in making their predictions.

We proposed CIS$^2$ as an alternative task to structure and evaluate language models for CCI. CIS$^2$ evaluation is a simplified, fairer measurement of CCI performance than BLEU. By finetuning a T5 model on our CIS$^2$ task, it correctly selects the causal statement 4.3% more than a model trained on the original GLUCOSE task. We note this is using heuristically converted CIS$^2$ labels, and collecting ground-truth CIS$^2$ labels for training would lead to even better performance.

Overall, we found that GLUCOSE indeed encodes contextual commonsense information, and T5 has capacity to learn this. Therefore, the challenge for future researchers is to leverage GLUCOSE and other contextual commonsense inference datasets' knowledge representations appropriately and avoid conflation of language generation.

---

[7]Experiments comparing CIS$^2$ to models that are trained to generate only specific rules can be found in Appendix A.6.

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

# A Appendix

## A.1 Acknowledgements

We thank the authors of GLUCOSE, in particular Or Biran and Lori Moon, for their helpful assistance in working with the GLUCOSE dataset and codebase. We also thank Daphne Ippolito and the anonymous reviewers for their comments and suggestions.

This material is based upon work supported by the National Science Foundation under Grant #2030859 to the Computing Research Association for the CIFellows Project.

## A.2 Ethical Considerations and Broader Impacts

The methods used in our paper build in large part upon work by prior researchers. The T5 (Raffel et al., 2020) language model we used was pre-trained on a massive dataset for many days. Despite the energy usage, T5 has proved be a valuable tool that can be used for countless downstream NLP applications, ours included. As for our own trained models, we note that we further fine-tuned T5 on an array of diagnostic and custom tasks. During development, we made sure to pilot any experiments on smaller datasets, and we carefully managed our GPU and CPU usage throughout.

As for the data used, the ROCStories (Mostafazadeh et al., 2016) and GLUCOSE (Mostafazadeh et al., 2020) datasets, in which our work builds on, involved a great deal of careful task design and interaction with crowd-source workers. We thank these researchers for their ethical treatment of their crowdsource workers, with fair pay and two-way communication (Moon et al., 2020).

We will publicly release all our code, from data preprocessing, to model training, to final evaluation, to ensure that our work is fully reproducible.

The broader impacts of our work outside its immediate subject are several. First, our work takes a step towards analyzing stories, which are something fundamentally human, and that machines have yet to master. Second, we have encouraged NLP researchers in general to think more carefully about the structure of a task, before defaulting to the latest state-of-the-art language model. For example, we found that our $CIS^2$ task, which is simpler and thus requires less training resources than the language generation task, performs better on capturing contextual commonsense inference.

## A.3 Reproducing Our Work

We make our code publicly available at a Github link. The codebase includes complete preprocessing, training, and evaluation scripts, to take the raw GLUCOSE CSVs and T5 checkpoints, and train both diagnostic and $CIS^2$ models. We will also release the final trained checkpoints.

We also include our code to reproduce the original GLUCOSE experiments. We model this closely to the original GLUCOSE paper, starting from their provided code repository.

## A.4 Reproduction Results

We report the results we obtained on the original GLUCOSE task in Table 5. We report per-dimension BLEU, as was done prior, as well as the weighted average BLEU across all dimensions. We find that the reported numbers from (Mostafazadeh et al., 2020) and their provided Tensorflow checkpoint are essentially consistent.

Our replication results (done with the `transformers` package (Wolf et al., 2019)) achieve 4-5 BLEU points lower, due to resource limitations and slight differences in experimental setup (i.e. we had far less GPU resources and and training time). For consistency's sake all of our experiments use the same setup as replicated t5-large (termed Original in the main text), and thus use this as the baseline.

We report results on the test set, but choose to evaluate BLEU on only the first of the three provided references for each test set entry. This is because the GLUCOSE train set only has one reference per entry, not 3, and we carved a small development set out of the train set, since no train/development split was provided. We evaluate our custom development and the original test set the same way, with 1 reference per entry.

## A.5 Training Setup and Hyperparameters

We trained our models on 2 NVIDIA Quadro RTX 6000 GPUs, with 24 GB vRAM each. We train up to 10 epochs, early stopping after 10 checkpoints without improvement on the validation set. Depending on the task, the models finish training between 6 to 34 hours. The GLUCOSE authors trained their model far more – for 72 hours on 8 TPUs – which can explain our lower BLEU scores.

We use the exact same hyperparameters as in Raffel et al. (2020), following Mostafazadeh et al. (2020), with one major exception: we use

| Model | Level | avg | 1 | 2 | 3 | 4 | 5 | 6 | 7 | 8 | 9 | 10 |
|---|---|---|---|---|---|---|---|---|---|---|---|---|
| (Mostafazadeh et al., 2020) | Specific | N/A | 72.5 | 73.8 | 70.5 | 81.1 | 71.7 | 73.9 | 79.3 | 80.2 | 86.6 | 66.9 |
| (Mostafazadeh et al., 2020) | General | N/A | 66.4 | 68.5 | 69.8 | 76.8 | 68.6 | 67.6 | 73.0 | 77.0 | 86.8 | 57.5 |
| GLUCOSE TF-checkpoint | Specific | 75.7 | 71.9 | 69.8 | 75.8 | 75.9 | 73.3 | 75.2 | 79.8 | 80.2 | 85.5 | 69.9 |
| GLUCOSE TF checkpoint | General | 70.1 | 66.4 | 66.4 | 70.1 | 72.1 | 70.0 | 69.2 | 71.6 | 72.4 | 82.0 | 61.0 |
| replicated t5-large | Specific | 70.7 | 65.9 | 60.4 | 63.8 | 76.5 | 69.0 | 66.7 | 72.6 | 74.0 | 82.4 | 76.0 |
| replicated t5-large | General | 66.2 | 61.3 | 59.9 | 60.4 | 68.8 | 61.3 | 60.5 | 65.0 | 68.1 | 75.8 | 80.4 |

Table 5: Test Set Results for the original GLUCOSE task. The first rows are the original results, the second are decoded by us using the provided GLUCOSE TF checkpoint, and the third are our best-effort replications.

| model | spec | sp1-5 | sp6-10 | gen | ge1-5 | ge6-10 |
|---|---|---|---|---|---|---|
| ORIGINAL | 70.7 | 67.1 | 74.4 | 66.2 | 62.3 | 70.0 |
| HISTORY | 35.9 | 36.9 | 34.9 | 50.4 | 50.1 | 50.7 |
| MASK X | 41.6 | 38.8 | 44.4 | 49.6 | 50.4 | 48.8 |
| HISTORY+X | 68.3 | 66.2 | 70.4 | 65.5 | 61.8 | 69.3 |
| ORIGINAL-SPEC | 67.6 | 60.5 | 74.8 | NA | NA | NA |
| HISTORY-SPEC | 37.6 | 36.1 | 39.0 | NA | NA | NA |
| MASK X-SPEC | 42.5 | 41.3 | 43.8 | NA | NA | NA |
| HISTORY+X-SPEC | 65.6 | 62.0 | 69.3 | NA | NA | NA |

Table 6: Test SacreBLEU scores for all tasks. The first 4 rows are the same as in Table 4—the models that outputted both specific and general rules. The last 4 rows are for models outputting specific rules only.

a learning rate of 1e-4 instead of 1e-3, which we found to converge too quickly.

### A.6 Specific-Only Results

Given that $CIS^2$ only considers the specific rule, one may ask how the GLUCOSE models trained to generate only specific rules would perform. We therefore train 4 "specific-only" models, one for each of the 4 diagnostic tasks of Section 4. We denote specific-only models with the suffix -SPEC and we compare the results to the specific+general models (as in the main text) without a suffix.

Table 6 compares the BLEU results, whereas Figure 4 compares the $CIS^2$ results. We see that the specific+general models and the specific-only models perform similarly. This confirms the findings of Mostafazadeh et al. (2020), where T5 can effectively learn both specific and general rules jointly. As both BLEU scores and $CIS^2$ classification accuracy are similar, we report the specific+general model results in the main paper to be consistent with prior work.

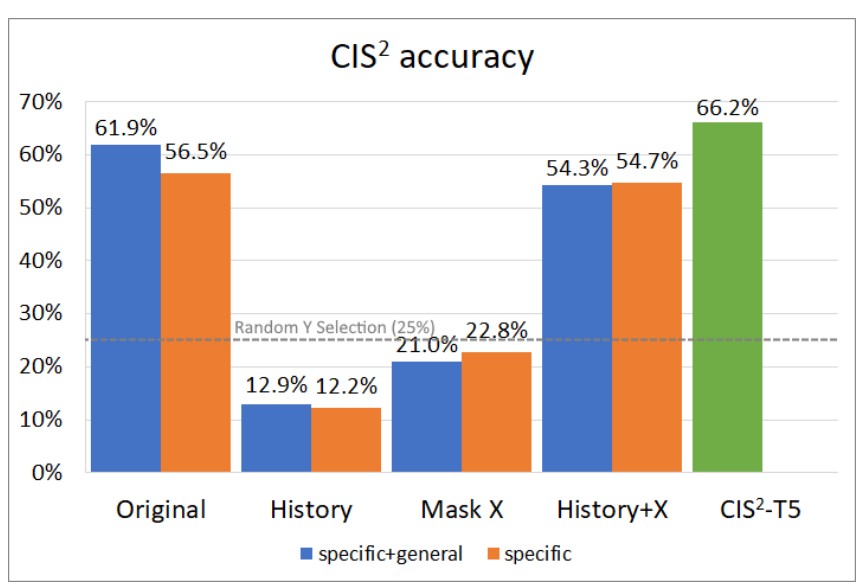

Figure 4: CIS² accuracy results, comparing specific+general models vs. specific-only models. The specific+general results are the same as in Figure 3.