# OpenReview forum: "$\text{CIS}^2$: A Simplified Commonsense Inference Evaluation for Story Prose"
_aclweb.org/ACL/2022/Workshop/CSRR — ACL 2022 Workshop CSRR_

### Official Review · Reviewer_EoNj · 2022-03-22
**The paper pinpointed a valid issue with the existing CCI task formulation. However, the findings are somewhat obvious, and the newly designed task might not generalize to settings where the CS inference is implicit and not part of the given story.**

**Rating:** 6
**Confidence:** 4

**Review:**

This paper critiques existing methods on Contextual Commonsense Inference (CCI), which conflates generation and reasoning tasks. The authors propose reframing the CCI task as a classification task (called Cis2) to isolate commonsense reasoning from the generation. This helps in evaluating the commonsense inference ability of a model irrespective of its generation performance. For this, they convert story sentences into output tags which avoids a partial match between input and output sequences. The model is then required to generate an abstracted output which contains story sentences' tags instead of full sequences.

Pros:
* It is important to evaluate the reasoning abilities of models in isolation from their generation abilities and the author pinpointed a valid issue with original GLUCOSE task formulation.


Cons:
* Most of the findings from the diagnostic tests are obvious and expected (see comments).
* Too much content is provided about Mostafazadeh et al. 2020 which could be easily skipped and referred to the original paper.
* It is not clear how the newly designed task formulation handles cases where the inference output Y is not explicitly stated in the given story.

Comments:
1- How does your task reformulation handle cases where the inference output Y is not explicitly stated in the story? As the authors mention in Line 196, the CS inference Y might or might not be part of the story. And example from the original GLUCOSE paper is:
“Gage wants safety” Causes/Enables “Gage turned his bike”, while “Gage wants safety” is never stated in the story and should be inferred thus can not be replaced by a tag from the story.

2- Line 259: while I agree with the authors that the original task formulation suffers from conflation of CCI and language generation tasks, I think this can be solved mostly by 1) removing the selected sentence X from the output, and 2) including a better evaluation metrics that accounts for semantic similarity such as BertScore.

3- Line 367-369: Isn't this obvious? if in the training data, output always copy/paraphrase X, it's expected that the model learns this pattern and consequently the BLEU score would be high. The issue is why the model should generate X in the first place. Without including X in the output (no matter if X is in the input or not) the evaluation using n-gram overlap would be less unreliable.

4- Line 380: In my opinion copying is an easier task.

5- The first 5.5 pages are allocated for background and related work and only on page 6 the author started to talk about their proposed task Cis2.

6- It is helpful to explicitly mention somewhere in the paper that you are using a generative classifier where the model GENERATES one of the 100 possible output sequences (using T5) and it’s not a 100-way classification task.

7- Line 449: For what portion of the original data could the authors find output Y explicitly mentioned in the input story? And why not discard those with very low similarity scores?

Typo:

Line 406: the The → The

Line 407: footnote after punctuation.

---

### Official Review · Reviewer_S91P · 2022-03-22
**Interesting hypothesis "Contextual Commonsense Inference should not be conflated with NLG as in GLUCOSE" but not much evidence.**

**Rating:** 4
**Confidence:** 4

**Review:**

What the paper is about: The paper argues that for contextual commonsense inference (commonsense understanding in some story), the GLUCOSE task conflates a different skill of natural language generation, which also brings in the ills of BLEU metrics. They propose the task of CIS2 which instead of asking the model to generate a commonsense inference, merely asks it to pick/classify the correct sentence prediction. They compare with different diagnostics/ablations of the original GLUCOSE task by removing parts of the input. They find that models trained on these ablations of GLUCOSE-Original perform worse than one trained on CIS2 (note that all these variants are based on the same GLUCOSE dataset) -- when evaluating on CIS2 metric of classification.

Key Shortcoming: There is no independent evidence that shows a classification task is better than generation task in training for "contextual commonsense inference." The only evidence is on the same metric of CIS2 which seems biased. The argument sounds like "Standardized Tests are not good benchmarks of creativity, so we propose instead teaching/testing students the skill of playing chess. We find that students preparing for different standardized tests are worse than students preparing for chess -- when evaluated on chess." The core hypothesis remains untested: whether chess playing (CIS2) is a better metric and task for creativity (commonsense contextual inference) than standardized testing (generation). Some ways that this could have been evaluated are:
1. human studies - do annotators find one model to exhibit more commonsense than others in some way?
2. independent downstream task - does the model trained on CIS2 outperform those trained on generation on some third task?

---

### Official Review · Reviewer_CzWL · 2022-03-23
**Insightful Analysis on CCI**

**Rating:** 7
**Confidence:** 3

**Review:**

The authors present analysis on contextual commonsense inference (CCI) using GLUCOSE, a story dataset annotated with commonsense explanations. They argue that the conflation of CCI with language generation (used in original GLUCOSE task) hinders model performance and the evaluation protocol also has issues. They propose to separate CCI from NLG by proposing CIS^2 and show improvement.

Strength:
- It's always good to see (and enjoyable to read!) analysis papers that tackle a previously studied problem from a new angle and provide insights. CIS^2 looks at the CCI problem and critiques a previous task formulation which I think provides important reflections for other researchers working on this problem.
- The experiment design is well-reasoned and thoroughly described. I really like the different diagnosis task settings trying to disentangle different factors that might influence CCI performance.
- The analysis on evaluation metrics also provides interesting insights

Places to improbe:
- I do not have major weaknesses to point out, but I think some parts of the writings can be much more concise. Especially Section 3 where the authors spent 2 pages on reviewing and introducing a previous work's data and task (I know it's crucial background but think could be shortened)
- Significance tests would be beneficial to be included for Table 4 and others.

---

### Decision · Program_Chairs · 2022-03-28

Accept